# Chronic Intermittent Hypoxia Exposure Alternative to Exercise Alleviates High-Fat-Diet-Induced Obesity and Fatty Liver

**DOI:** 10.3390/ijms23095209

**Published:** 2022-05-06

**Authors:** Yunfei Luo, Qiongfeng Chen, Junrong Zou, Jingjing Fan, Yuanjun Li, Zhijun Luo

**Affiliations:** 1Jiangxi Provincial Key Laboratory of Tumor Pathogens and Molecular Pathology, Department of Pathophysiology, Schools of Basic Sciences, Nanchang University, Nanchang 330031, China; 356535517027@email.ncu.edu.cn (Y.L.); chenqiongfeng@ncu.edu.cn (Q.C.); ydzjr@gmu.edu.cn (J.Z.); 401442719007@email.ncu.edu.cn (J.F.); 401441220005@email.ncu.edu.cn (Y.L.); 2Queen Mary School, Nanchang University, Nanchang 330031, China

**Keywords:** chronic intermittent hypoxia, obesity, fatty liver, epinephrine, AMPK, inflammation

## Abstract

Obesity often concurs with nonalcoholic fatty liver disease (NAFLD), both of which are detrimental to human health. Thus far, exercise appears to be an effective treatment approach. However, its effects cannot last long and, moreover, it is difficult to achieve for many obese people. Thus, it is necessary to look into alternative remedies. The present study explored a noninvasive, easy, tolerable physical alternative. In our experiment, C57BL/6 mice were fed with a high-fat diet (HFD) to induce overweight/obesity and were exposed to 10% oxygen for one hour every day. We found that hypoxia exerted protective effects. First, it offset HFD-induced bodyweight gain and insulin resistance. Secondly, hypoxia reversed the HFD-induced enlargement of white and brown adipocytes and fatty liver, and protected liver function. Thirdly, HFD downregulated the expression of genes required for lipolysis and thermogenesis, such as *UCP1*, *ADR3(beta3-adrenergic receptor)*, *CPT1A*, *ATGL*, *PPARα*, and *PGC1α*, M2 macrophage markers arginase and *CD206* in the liver, and UCP1 and PPARγ in brown fat, while these molecules were upregulated by hypoxia. Furthermore, hypoxia induced the activation of AMPK, an energy sensing enzyme. Fourthly, our results showed that hypoxia increased serum levels of epinephrine. Indeed, the effects of hypoxia on bodyweight, fatty liver, and associated changes in gene expression ever tested were reproduced by injection of epinephrine and prevented by propranolol at varying degrees. Altogether, our data suggest that hypoxia triggers stress responses where epinephrine plays important roles. Therefore, our study sheds light on the hope to use hypoxia to treat the daunting disorders, obesity and NAFLD.

## 1. Introduction

Obesity arises from an abnormal and excessive accumulation of fat, leading to the impairment of health. From 1975 to 2016, the prevalence of obesity almost tripled, with at least 2.8 million related deaths each year [1]. Obesity is an important risk factor for metabolic syndrome and comorbidities such as type 2 diabetes (T2DM), nonalcoholic fatty liver (NAFLD), hypertension, obstructive sleep apnea, myocardial infarction (CAD), chronic kidney disease, osteoarthritis, and malignancies, leading to increased mortality in obese people [2]. Nowadays, obesity continuously increases the burdens of the healthcare systems. It is estimated that obesity accounts for 0.7–2.8% of the total healthcare costs in developed countries and the average costs for obese people are 30% higher than those of normal-weight individuals [3]. Due to the deficiency of effective treatment of its commodities, mitigation of obesity is the primary measure to prevent its harmfulness to human health.

In general, the balance between energy intake and expenditure is a determining factor of bodyweight. Overeating or reduced energy expenditure can disrupt the balance, leading to overweight and obesity. Hence, diet control and adequate amounts of exercise are effective methods to control bodyweight. However, the duration of the effect is limited because the bodyweight often tends to go back after reaching a plateau [4,5]. Furthermore, diet control and exercise are difficult to achieve for many obese people.

The prevalence and severity of NAFLD is usually paralleled to degrees of obesity [3]. NAFLD, which progresses from simple steatosis (SS) to nonalcoholic steatohepatitis (NASH), may advance to cirrhosis and hepatocellular carcinoma (HCC) [3], where inflammation plays an important role. While the prevalence of NAFLD is 25–30% in the general population, it markedly increases in the obese population, which rises up to 90% in morbidly obese patients [3]. Likewise, NAFLD-related mortality continues to increase, resulting from cirrhosis, chronic kidney disease, cardiovascular disease, and malignancies [3]. In many circumstances, alleviation of obesity also improves NAFLD.

Exposure to hypoxia can be an effective alternative to exercise but is an understudied area to control bodyweight. Epidemiological studies have reported that people living in high-altitude areas have a lower prevalence of obesity and diabetes [6,7]. The main environmental differences between high-altitude and plain areas are temperature, oxygen concentration, and baric pressure [8,9]. As a result, it is impossible to dissect a single causal factor in the case of high-altitude habitats. For instance, cold stimulation could also alleviate obesity [10]. Yet, the study of oxygen concentration as a single factor is needed to define its roles in obesity and associated disorders. Many investigations of hypoxia have been conducted and both harmful and beneficial effects have been reported [11]. The underlying reason might be attributable to different methods of hypoxia (time, oxygen concentration, hypoxia mode, etc.) [11].

The hypoxic condition applied in the present study is to place mice under 10% oxygen for 1 h every day. This method is different from high-altitude hypoxia and would not cause sickness similarly to exposure to high altitude. We found that the exposure of mice to the low oxygen concentration conferred resistance to the high-fat-diet (HFD)-induced gain in bodyweight and remission of the severity of fatty liver. Hypoxia induced the activation of AMPK, the upregulation of UCP1 and other molecules required for lipolysis and fatty acid oxidation, and the induction of anti-inflammatory M2 macrophage marker CD206 in the liver. Furthermore, hypoxia increased the levels of serum epinephrine, and the administration of epinephrine ameliorated HFD-induced bodyweight gain and NAFLD, while many effects of hypoxia were prevented by propranolol to varying degrees.

## 2. Results

### 2.1. Chronic Intermittent Hypoxia Reduced Overweight/Obesity in Mice and Improved Glucose Tolerance

According to the Altitude and Oxygen Chart (https://www.higherpeak.com/altitudechart.html, accessed on 18 April 2022) [12], we chose 10% O_2_ (~50% normoxia) in this study, which is equivalent to 5791 m (approximately equal to the altitude of Kilimanjaro). The condition we used is isobaric, which is easily tolerable as compared to Kilimanjaro where it is hypobaric. To assess the effect of chronic intermittent hypoxia on bodyweight, mice were placed into an hypoxia chamber and treated with 10% O_2_ (defined as hypoxia in present work) controlled by gas nitrogen for 1 h every day or with ambient oxygen (normoxia) as a control. They were divided into five groups (8 for each), as shown in the scheme (Figure 1A): (1) chow diet under normoxia; (2) chow diet under hypoxia; (3) HFD under normoxia; (4) after feeding with chow diet for 6 weeks, the mice were switched to the HFD and simultaneously subjected to hypoxia; and (5) after feeding with chow diet for 6 weeks, followed by HFD for 4 weeks, the mice were then continuously fed with the HFD and subjected to hypoxia. Experiments were ended at 12 weeks (age of 18 weeks). As shown in Figure 1B,C, the bodyweight of mice fed with the HFD under normoxia gradually increased from 23.43 ± 0.3 g to 48.90 ± 1.8 g at the 12th week, and hypoxia significantly conferred resistance to the HFD, regardless of whether hypoxia started at the same time as HFD feeding (BW from 24.40 ± 0.5 g to 40.65 ± 1.4 g) or at the 4 weeks of HFD (BW from 21.70 ± 0.6 g to 38.40 ± 1.5 g) (in these two scenarios, *p* < 0.001). As a control, the same hypoxic treatment did not appear to affect the bodyweight on mice with chow diet feeding. Changes in bodyweight under the HFD were not attributed to decreases in appetite, as average food intakes were not affected by low oxygen regardless of the fact that the food intake was less with the HFD than that with the chow diet (Figure 1D).

We then measured glucose profiles in response to hypoxia treatment. It was found that the fasting blood glucose of the HFD-induced mice under normoxia reached 9.33 ± 0.24 mmol/L, while simultaneous administration of hypoxia gave rise to resistance to HFD (6.77 ± 0.2 mmol/L, *p* < 0.001). The mice fed with HFD for 4 weeks before hypoxia treatment showed a declining trend of fasting blood glucose but without significance (Figure 1E). Next, the GTT assay showed that the HFD dampened glucose tolerance, as compared with the chow diet (*p* < 0.0001). The impaired GTT was ameliorated by both simultaneous and subsequent treatment with hypoxia (*p* = 0.0003, *p* = 0.00226, respectively) (Figure 1F). The ITT assay also showed an improvement in insulin tolerance by hypoxia (*p* < 0.05) (Figure 1G). These results suggest that hypoxia could be an effective approach to counteract HFD-induced obesity and associated abnormalities of glucose metabolism. To ascertain if hypoxia induces damages to tissues, we performed H&E staining of the heart, lung, and kidney tissues and did not find any structural changes in morphology (see Appendix A).

### 2.2. Chronic Intermittent Hypoxia Reduced the Fat Content of Mice and Promoted the Expression of Thermogenic Genes in Brown Fat

To assess the effect of hypoxia on fat distribution, adipose tissues from different locations were first collected and weighed. As shown in Figure 2A–C, the contents of subcutaneous fat, perirenal fat, and brown fat in the mice fed with the HFD were greatly increased as compared to the chow diet group, and significantly reduced by hypoxia treatment (*p* < 0.001); however, no evident changes in these tissues were observed in the mice fed with the chow diet between normoxia and hypoxia. The brown fat tends to transit to white fat (Figure 2D). We then examined the morphology of white fat and brown fat. Thus, under normoxic conditions, the size of adipocytes from white adipose of the mice fed with HFD was remarkably larger than that with the chow diet, whereas under hypoxia, the size of adipocytes in the HFD group in general was similar to that of the chow diet group (Figure 2D). Vacuoles were obviously larger in the brown fat of the HFD group than those treated with hypoxia.

We performed immunochemical staining of the brown fat tissues with antibodies against PPARγ and UCP1, two parameters for brown fat thermogenesis. As shown in Figure 2E–G, our results revealed that HFD decreased the expression of UCP1 (*p* < 0.05) and PPARγ (*p* < 0.001), as compared to chow diet. In contrast, hypoxia greatly upregulated the expression of UCP1 (*t*-test, *p* < 0.05) and PPARγ (*p* < 0.001). These results suggest that hypoxia increases thermogenesis.

### 2.3. Chronic Intermittent Hypoxia Alleviated Fatty Liver and Altered Gene Expression

We dissected livers of the experimental mice and found that the livers from the HFD-fed mice under normoxia were much heavier than those of the chow diet group or the HFD group treated with hypoxia (Figure 3A). The liver under HFD-normoxia appeared pink, while hypoxia turned the color to dark red, close to that of the chow diet group (Figure 3B). H&E staining showed small vacuoles and scattered ectopic fat in the liver of the HFD-fed mice, which were markedly reduced by hypoxia. Further, Oil Red O stained many more lipid droplets in the liver of the HFD mice under normoxia than those under hypoxia.

To dissect molecular changes associated with fatty liver, we performed Western blot and qPCR to examine changes in gene expression related to glucose and lipid metabolisms, and markers of M2 macrophages in the liver. As shown in Figure 3C,D, the protein level of UCP1 was greatly downregulated in the HFD-fed mice, while it was recovered by hypoxia. No significant changes in FASN protein levels were observed in the HFD mice between hypoxia and normoxia, although they were reduced on the HFD relative to those on the chow diet (Figure 3C,E). The qPCR results revealed that hypoxia reduced the mRNA level of *SCD1* in the HFD group, compared with the normoxic environment (*t*-test, *p* < 0.05) (Figure 3F). In addition, hypoxia significantly increased mRNA levels of genes in the HFD mice closely related to fatty acid oxidation, lipolysis, and thermogenesis including *PGC1α*, *ATGL*, *PPARα*, *and ADR3*, as well as *UCP1* (*t*-test, *p* < 0.05). The *CPT1A* mRNA was increased by hypoxia, but not significant (Figure 3F).

A statistical decrease in arginase was observed in the HFD-fed mice under normoxia, as compared with chow diet feeding (*t*-test, *p* < 0.05), while hypoxia resulted in a trend of elevation, albeit no significance. *CD206* expression levels exhibited similar changes (Figure 3G).

### 2.4. The Role of Epinephrine in Hypoxia-Induced Physiological Changes

As hypoxia might trigger a stress response similarly to exercise, we asked whether epinephrine was involved in mediating the effect of hypoxia. We first placed mice in an hypoxic chamber for one hour and measured the levels of serum epinephrine at different time points (from 0 to 24 h). Figure 4A shows the time course changes in the serum levels of epinephrine, which started to increase immediately after hypoxia, reaching a maximum at 2 h, and declined thereafter to a level that was still higher than that prior to hypoxia (*p* < 0.05). In contrast, epinephrine levels remained constant under normoxia within the first two hours, although changes afterward were not examined, along with circadian rhythm. At the end of 24 h, the level of epinephrine was significantly higher under hypoxia than normoxia despite declining (*p* < 0.05).

We then assessed if epinephrine mediated the effect of hypoxia. Thus, animals were grouped as follows (Figure 4B): (1) chow diet under normoxia, (2) HFD under normoxia, (3) HFD first for 4 weeks and then injected intraperitoneally with epinephrine, (4) HFD under normoxia for 4 weeks and then subjected to hypoxia, (5) HFD under normoxia for 4 weeks and then subjected to hypoxia along with i.p. injection of propranolol. From the beginning of HFD feeding, bodyweight was measured every two weeks and the increment recorded above the starting point as shown in Figure 4C. The experiments were ended at 12 weeks (84 days after HFD feeding).

Changes in the trend of bodyweight in mice are shown in Figure 4C. Increases in bodyweight of HFD-fed mice under normoxic conditions were significantly greater than those under hypoxia or injected with epinephrine (*p* < 0.001), while the HFD-fed mice injected intraperitoneally with propranolol conferred moderate resistance to hypoxia, but without significance. Likewise, at the end point, the increase in bodyweight gain of the HFD mice injected with epinephrine was similar to that of mice under hypoxia treatment, both of which were significantly greater than that of the HFD-fed mice under normoxia (*p* < 0.01) (Figure 4D). However, the injection of propranolol partially inhibited the effect of hypoxia without significance (*p* = 0.162), suggesting that part of the hypoxia effect on bodyweight is mediated by epinephrine. Even at the end of the experiment, we still observed that the endogenous level of epinephrine was higher under hypoxia treatment than those of lean and obese mice under normoxia (*p* < 0.05) (Figure 4E).

We next examined fasting blood glucose levels and found that chronic treatment with epinephrine offset the HFD-induced increase in fasting blood glucose levels (*p* < 0.05), although it did not completely restore to the levels fed on the chow diet (Figure 4F). Our results also showed that the improvement of fasting glucose levels by hypoxia was prevented by propranolol (*p* < 0.01). Finally, we used ALT and AST as parameters to assess if liver function was affected by hypoxia. Our data revealed that serum levels of ALT and AST were much greater in HFD-fed mice than those of the chow diet group (Figure 4G–H). Compared to the HFD-fed mice under the normoxia, hypoxia or injection with epinephrine alleviated liver damage, while injection of propranolol in the hypoxia-treated HFD mice modestly offset the hypoxia effect (Figure 4H). These results suggest that hypoxia-induced protection of the liver was not completely mediated by epinephrine, although the latter alone could confer the protection. An alternative interpretation is that the dose of propranolol used in the study might not be sufficient to antagonize the effect of increased endogenous epinephrine.

### 2.5. Epinephrine Mediated the Effect of Hypoxia on Fat Deposits in Liver and Adipose Tissues and Associated Molecular Changes

We performed H&E staining of white fat and brown fat and observed that HFD feeding enlarged the size of adipocytes, which was mitigated by the injection of epinephrine or hypoxia, whereas the effect of hypoxia was reversed to some degree by the injection of propranolol (Figure 5A). The liver appearance of HFD-fed mice was pink, while the color turned to dark red closer to that of the chow diet by injection of epinephrine or hypoxia treatment (Figure 5B). In parallel, the microstructure was assessed by H&E and Oil Red O staining. The results showed that HFD induced larger lipid droplets, which were reduced by epinephrine and hypoxia. The change caused by hypoxia was obviously prevented by propranolol injection.

The content of triglycerides and cholesterol was assayed in the liver tissue (Figure 5C,D). It was found that the amount of hepatic triglycerides and cholesterol in the HFD-fed mice was much greater than that of chow diet-fed mice (*p* < 0.05, *p* < 0.001, respectively), whereas epinephrine and hypoxia significantly blunted the increase induced by the HFD (*p* < 0.05, *p* < 0.01, respectively). The injection of propranolol diminished the beneficial effects of hypoxia (*p* < 0.05).

To further explore the mechanisms underlying the changes in lipid and glucose metabolisms, we selected a few related molecules (Figure 6). First, our Western blot data showed that hypoxia treatment of HFD-fed mice upregulated hepatic levels of UCP1, p-AMPK, PKAc, PGC1α, and ADR3 (*p* < 0.05), which were mimicked by the injection of epinephrine (*p* < 0.05) and counteracted by propranolol (*p* < 0.05 except for PGC1α); second, we checked the expression of CPT1A by IHC staining of the liver tissue and found that CPT1A expression was reduced in the liver of obese mice, which was restored by hypoxia and epinephrine (*p* < 0.05), while the injection of propranolol prevented the rise triggered by hypoxia (*p* < 0.05) (Figure 6G,H); third, we assessed the protein expression of M2 macrophage marker CD206 in the liver of obese mice in response to hypoxia, epinephrine, and propranolol. As shown in the Western blot (Figure 6I,J), the expression was high in the chow-diet-fed mice and diminished in the HFD-induced obese mice, which was restored by epinephrine (*p* < 0.05) and hypoxia (*p* < 0.05). Injection of propranolol prevented the effect of hypoxia. Finally, we measured the levels of mRNA for *ATGL*, *UCP2*, *C/EBP*, *adiponectin*, and *PPARα* (Figure 6K). We found that these mRNAs were suppressed in obese mice to some extent and increased by epinephrine and hypoxia. Whatever the observed effects of hypoxia were, they were blocked by propranolol. Altogether, our data support that hypoxia stimulates the gene expression necessary for fatty acid oxidation, lipolysis, and thermogenesis to overcome the detrimental effects of obesity and ameliorates NAFLD.

## 3. Discussion

Obesity often concurs with NAFLD [13]. Thus far, diet control and exercise appear to be effective approaches to alleviate these disorders. However, they are difficult to achieve for many obese people. Thus, the present study explores a noninvasive and easily tolerable physical approach that is hypoxia in the treatment of obesity. In this work, we employed HFD to induce overweight/obesity in C57BL/6 mice and applied 10% oxygen for one hour every day. We found that hypoxia exerted protective effects. First, hypoxia suppressed HFD-induced bodyweight gain and insulin resistance. Secondly, hypoxia reversed the HFD-induced enlargement of white and brown adipocytes (brown whitening) and fatty liver, and protected liver function. Thirdly, HFD caused the downregulation of modulators required for fatty acid oxidation, lipolysis, and thermogenesis, such as UCP1, ADR3, CPT1A, ATGL, PPARα, and PGC1α, the M2 macrophage markers arginase and CD206 in the liver, and UCP1 and PPARγ in brown fat, while these molecules were recovered by hypoxia. Furthermore, hypoxia induced the activation of AMPK, an energy sensing enzyme. Fourthly, our results showed that hypoxia increased serum levels of epinephrine. Indeed, the effects of hypoxia were reproduced by the injection of epinephrine and suppressed by propranolol to varying degrees. Our findings are reminiscent of a previous report in a human study showing that exposure to hypobaric hypoxia caused bodyweight reduction in obese subjects [14].

Intermittent hypoxia training (IHT) was first employed for altitude preacclimatization and its concept was developed by scientists in the previous Soviet Union in the 1930s [15]. In addition to altitude preacclimatization, IHT has been explored for the treatment of a variety of clinical disorders, such as chronic lung diseases and bronchial asthma [16], cardiovascular diseases [17], diabetes mellitus [18], and Parkinson’s disease [19]. On the contrary, obstructive sleep apnea (OSA), another type of intermittent hypoxia with a shorter cycling of hypoxia and normoxia, yields opposite outcomes associated with many diseases such as diabetes, cardiovascular disease, and cancer [20]. The major differences of these two types of intermittent hypoxia are that (1) the IH of OSA occurs during sleeping while IHT occurs during awakening, and (2) the hypoxia-reoxygenation of OSA typically occurs for less than 60 s in duration while IHT lasts from minutes to hours. These differences lead to opposite responses, such as ventilation, arterial pressure, reactive oxygen species, and inflammation status.

It has been reported that hypoxia decreases blood levels of glucose, cholesterol, and basal Leptin, results in fat loss, and prevents steatosis in obese mice [14,21,22]. Previous studies have also documented that acute hypoxic training improves glucose tolerance and increases peripheral insulin sensitivity [23,24]. In a human study, intermittent hypoxia exposure has been shown to increase plasma epinephrine concentration and to improve insulin sensitivity [21]. An additional study has revealed that intermittent hypoxia alone or in combination with exercise increases glucose transporter-4 levels [25].

Similarly to exercise [26], acute intermittent hypoxia has been shown to increase blood levels of epinephrine [27]. The increased epinephrine can activate the adrenergic receptor β3 (ADR3) in the liver and adipose tissues [28], which then activates protein kinase A (PKA) and AMPK, resulting in the inhibition of lipogenesis and enhancement of lipolysis and fatty acid β-oxidation [29,30,31]. In addition, AMPK activation can promote mitochondria genesis by upregulation of PGC-1α [32]. In addition, activated AMPK enhances the expression of CD206, a hallmark of M2 macrophages [33], which leads to the transformation of macrophages from the pro-inflammatory M1 type to anti-inflammatory M2 type [34]. All of these integral effects may account for the inhibition of NAFLD. Interestingly, a recent study of Garcia et al. [35] showed that the liver-specific expression of transgene encoding active truncated mutant of α1 subunit of AMPK prevents hepatic steatosis in mice induced by HFD.

UCP1 can be activated in brown adipose tissue (BAT) upon cold exposure and norepinephrine (NA), both of which could cause the activation of AMPK [36,37]. In addition to brown fat, UCP1 is implicated to have a role in dissipating excess energy stored in the liver [37,38]. For example, the hepatic expression of UCPI leads to increased energy expenditure, decreased bodyweight, and reduced fat in the liver [37]. In contrast, the knockout of UCP1 induces fatty liver in HFD-fed mice [38]. Our results revealed that UCP1 was diminished in the liver of HFD-fed mice, which was reversed by hypoxia and epinephrine. It is possible that AMPK mediates the effect of hypoxia. In line with this, AMPK has been sown to regulate thermogenesis through UCP1-dependent and -independent mechanisms in adipocytes and the liver [39,40].

Our data revealed that many of these effects of hypoxia were mediated by epinephrine as they were imitated by epinephrine and suppressed by propranolol. Our observation seemed paradoxical with regard to the role of epinephrine in blood glucose as it is a hyperglycemic hormone. A possible explanation is that previous studies looked into the acute effect, while ours assessed the chronic effect of hypoxia, where its action through molecules such as AMPK and SCD1 [41] may prevail, resulting in an improvement in glucose profile and insulin resistance. Therefore, it is conceivable that epinephrine induces bodyweight loss because of its integral role in regulating lipolysis, lipogenesis, and thermogenesis [42] mainly through the activation of PKA and AMPK.

Of note, the role of HIF-1α in biological events is always considered whenever hypoxia is encountered. However, it is controversial regarding the role of HIF-1α in the regulation of energy homeostasis under hypoxia in whole body levels. On the one hand, HIF-1α is upregulated under hypoxic conditions, leading to the increased transcription of genes required for glucose utilization (e.g., Glut1, Glut4, phospho-fructose kinase, hexokinase, and lactate dehydrogenase, etc.) [43]. Thus, intermittent hypoxic training or exercise in high-altitude hypoxia increases the expression of HIF-1α [44,45], suggesting that HIF-1α is beneficial to the adaptation to hypoxia by the upregulation of gene expression for glycose utilization. On the other hand, genetic studies by the hepatic deletion of HIF-1α alleles in mice that modeled obstructive sleep apnea revealed that HIF-1α contributes to insulin resistance and NAFLD [46]. It has been shown that AMPK could positively or negatively regulate the expression of HIF-1α depending on biological context [47,48]. In future studies, we will assess changes in the expression of HIF-1α in our setting and, if any are observed, we will vigorously assess the role of HIF-1α in the amelioration of NAFLD by hypoxia.

In essence, the treatment of obesity and NAFLD is a daunting task. Even though diet control and exercise manifest some effects, these approaches are difficult to achieve for many obese people and the duration of the effect is limited. Our present study employs a noninvasive approach, a chronic intermittent hypoxia exposure alternative exercise to control bodyweight gain, glucose and lipid homeostasis, and the development of NAFLD. In light of our findings and recent reports that intermittent hypoxia stimulates the activation of AMPK [49,50], we propose a model in which the axis consisting of catecholamine-PKA-AMPK-effectors elicits biological events in response to chronic intermittent hypoxia. Finally, we should point out that our method of hypoxia is easy to operate relative to those using short cycling of hypoxia and normoxia or long-lasting hypoxia, which is not practical in terms of translational application. Therefore, our study sheds light on the treatment of obesity and NAFLD.

## 4. Materials and Methods

### 4.1. Chemicals and Reagents

Trizol reagents (Catalog number: DP419) for the total RNA preparation were purchased from Tiangen Biotech (Beijing, China). PrimeScript™ RT (RR047A) for reverse transcription was from TaKaRa Bio (Shanghai, China). SYBR Green kit (MF013-01) was from Mei5bio (Beijing, China). The BCA kit (PC0020), total cholesterol assay kit (BC1980), H&E (G1120), and Oil Red O (G1260) were from Beijing Solarbio Science and Technology Company (Beijing, China). The ELISA kit for epinephrine (KA1882) was from Abnova (Taiwan, China). Kits for alanine aminotransferase (ALT) (C009-2) and aspartate aminotransferase (AST) (C010-2) were from Nanjing Jiancheng Bioengineering Institute (Nanjing, Jiangsu, China). The kit for the triacylglycerol assay (E1013) was from Applygen (Beijing, China). Antibodies against p-AMPK (#50081), AMPK (#5831), and GAPDH (#5174T) were from Cell Signaling Technology (Danvers, MA, USA). Antibodies against UCP1 (ab135372), PPARγ (ab59256), CPT1A (ab128568), CD206 (ab64693), PGC1α (ab54481), PKAc (ab76238), and ADRβ3 (ab94506) were from Abcam (Cambridge, MA, USA). Antibody against FASN (#610963) was from BD Biosciences (Franklin Lakes, NJ, USA). Horse radish peroxidase-conjugated second antibodies were from Zsbio (Beijing, China). The high-fat diet (60% kcal/fat, 20% kJ/carbohydrate, 20% kJ/protein) (D12492) was from Research Diets Inc. (New Brunswick, NJ, USA), and the regular chow diet was from Keao Xieli Feed Cooperation (SPF-level mice maintaining chow) (Beijing, China).

### 4.2. Animal Model

C57BL/6 mice (male, 5 weeks old, SPF grade) were purchased from the Model Animal Research Center of Nanjing University. The mice were housed in the IVC environment at 23 ± 2 °C and a 12 h light-dark cycle. The animal protocol was approved by the Animal Care and Use Committee of Jiangxi Medical College, Nanchang University (Protocol number: NCDXSYDWFL-2015097).

For the hypoxia study, mice were placed into an hypoxia chamber and 10% oxygen was applied with nitrogen gas using Oxygen Controllers manufactured by BioSpherix (Parish, NY, USA). The mice were randomly divided into groups (8 per group) as indicated in the Results. One week after arrival at the animal center, the mice were fed with a high-fat diet or chow diet and treated with 10% oxygen for 1 h every day from the beginning of the HFD or at 4 weeks of HFD until sacrifice. The control groups were treated with ambient oxygen (normoxia) in the same chamber. Bodyweight and food intake were recorded weekly. To assess the effects of epinephrine, mice were randomly divided into different groups, as indicated in the Results. Whenever needed, intraperitoneal injection of epinephrine (0.1 mg/kg/day) or propranolol (2 mg/kg/day) was conducted. At the endpoint, the mice were sacrificed by cervical dislocation. The serum, liver, and adipose were collected and stored at −80 °C until experiments.

### 4.3. Glucose Tolerance Test (GTT) and Insulin Tolerance Test (ITT)

The assays were performed according to Regué et al. [51] In brief, at the 12th week, mice were fasted for 12 h but with free access to water and intraperitoneally injected with glucose (2 g/kg). Blood glucose was then determined for GTT using glucose strips at 0, 30, 60, 90, and 120 min. For ITT assay, one week after GTT, the mice were fasted for 4 h and intraperitoneally injected with insulin (1 IU/kg). Blood glucose was then determined using glucose strips at 0, 15, 30, 60, and 90 min.

### 4.4. Serum Epinephrine and Enzyme Assays

At the indicated time points, blood samples were collected from orbits of the experimental mice under anesthesia, clotted on ice for 10 min. Serum was collected after centrifugation at 3000× *g* rpm. Epinephrine was measured using the ELISA kit according to the manufacturer’s instruction. Serum concentrations of alanine aminotransferase and aspartate aminotransferase were determined following the manufacturer’s instruction.

### 4.5. Tissue Homogenization

The liver tissues (approximately 100 mg) were cut into small pieces in PBS to wash blood and centrifuged at low speed to discard the liquid. The pellets were then quick-frozen in liquid nitrogen and grinded in protein extraction buffer (50 mM Tris-Cl, pH 7.8, 25 mM β-glycerophosphate, 1 mM Sodium Vanadate, 1 mM EDTA, 1 mM EGTA, 1 mM DTT, 1% NP-40, protease inhibitor cocktails). The samples were incubated on a vertical mixer at 4 °C for 30 min, centrifuged at 12,000× *g* rpm in a microcentrifuge at 4 °C for 15 min, and the supernatant was collected in a new Ependorf tube. Protein contents were quantified using the BCA kit.

### 4.6. Measurement of Triacylglycerol and Total Cholesterol in Liver

The assays were performed according to protocols provided by the manufacturer (Beijing Solarbio Science and Technology Company, Beijing, China).

### 4.7. Western Blot

Western blot analysis was conducted as described previously [52]. Briefly, the same amounts of samples (20 μg) were subjected to SDS–PAGE (10%) and electrophoretically transferred to PVDF membranes, which were then blocked with 5% non-fat milk in Tris-buffered saline and 0.1% tween-20 (TBST) for 1–2 h and incubated in the primary antibody 1% BSA-TBST at 4 °C overnight. After washing with TBST three times, the membranes were incubated with a secondary antibody in 5% non-fat milk-TBST for 1–2 h at room temperature. Specific protein bands were visualized by ECL and quantified using Image J software if necessary.

### 4.8. Histochemistry

The liver or adipose tissues were fixed with 4% paraformaldehyde and embedded in paraffin. When needed, the paraffin-embedded tissues were sectioned to 5-micron slides, deparaffinized, and stained with hematoxylin and eosin (H&E staining). Immunohistochemical staining was performed according to the protocol of Abcam. In brief, the primary antibodies, UCP1 (1:500), PPARγ (1:200), FASN (dilution of 1:500), CPT1A (1:200), and CD206 (1:100), were incubated, followed by the horse radish hydroxylase-conjugated secondary antibody from Zsbio. The cryosections of the liver tissues (10 μm thickness) were stained with Oil Red O.

### 4.9. Quantitative Reverse Transcription–Polymerase Chain Reaction (RT–PCR)

All primers were synthesized by Tsingke Biotechnology Co., Ltd. (Beijing, China), according to the PrimerBank (https://pga.mgh.harvard.edu/primerbank/, accessed on 30 April 2022) and are listed in Appendix A. Total RNA was isolated from fresh-frozen liver tissues using Trizol reagents and qPCR was performed using the PrimeScript™ RT reagent kit according to the protocols provided by manufacturers. Each sample was amplified in triplicates and normalized with *18S rRNA* signals. Results were assessed by the comparative threshold cycle value method (2 − ΔΔ Ct) for relative quantification of gene expression.

### 4.10. Statistical Analysis

Quantitative data were analyzed using GraphPad Prism 5 software (San Diego, CA, USA). Data are presented as mean ± SEM and analyzed using Student’s *t*-test, and one- or two-way ANOVA with the Sidak post hoc test. *p* < 0.05 was considered a significant difference.

## Figures and Tables

**Figure 1 ijms-23-05209-f001:**
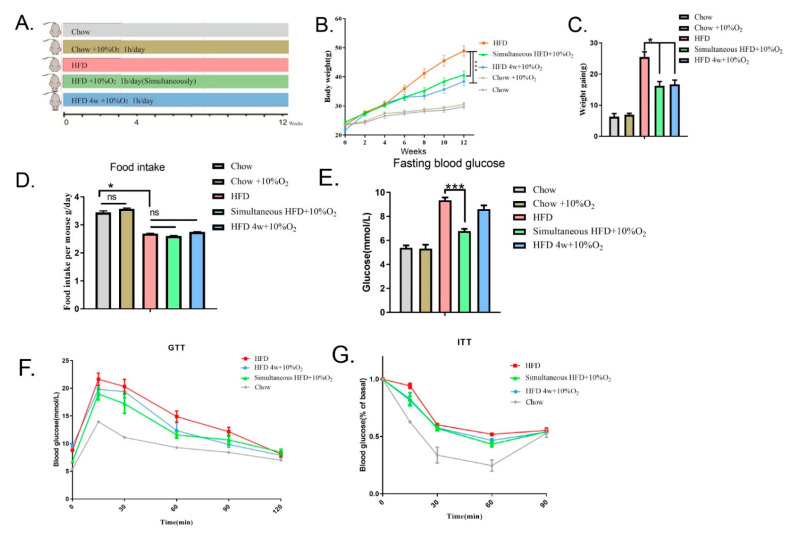
Chronic intermittent hypoxia reduces the bodyweight gain induced by HFD and improves insulin resistance. Experimental mice were exposed to ambient oxygen (normoxia) or 10% oxygen for one hour every day. (**A**) Scheme of grouping (8 mice per group): (1) chow diet and normoxia, (2) chow diet exposed to 10% oxygen, (3) HFD and normoxia, (4) HFD and 10% oxygen simultaneously, (5) HFD for 4 weeks and then 10% oxygen applied. (**B**) Body weight was measured every two weeks. Statistical analysis was performed by two-way ANOVA (mean ± SEM, *n* = 8). *** *p* < 0.005. (**C**) The value of bodyweight gain was obtained by subtracting bodyweight at the starting point from the endpoint (mean ± SEM). * *p* < 0.05. (**D**) Food intake (g/day/mouse) was recorded for one week and averages were presented. * *p* < 0.05, ns means not significant. (**E**) Fasting blood glucose levels wee measured. Significance was tested by Student’s *t*-test (mean ± SEM), *** *p* < 0.005. (**F**,**G**) Glucose tolerance test (GTT) and insulin tolerance test (ITT) were carried out. AUC for F: HFD, 1781; HFD 4 w + 10% O_2_, 1590; Simultaneous HFD + 10% O_2_, 1517; Chow, 1136. AUC for G: AUC: HFD, 59.05; HFD 4 w + 10% O_2_, 54.77; Simultaneous HFD + 10% O_2_, 53.85; Chow, 39.82. Significance was tested using two-way ANOVA (mean ± SEM). *p* < 0.05 (Group 3 vs. Group 4 and Group 3 vs. Group 5, respectively).

**Figure 2 ijms-23-05209-f002:**
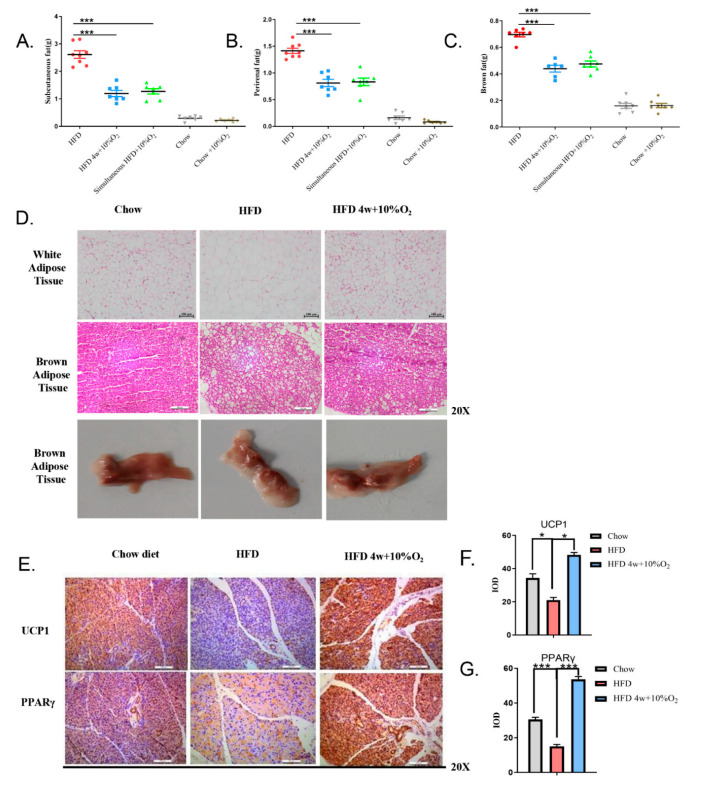
Hypoxia reduces the fat content of mice and promotes the expression of thermogenic genes in brown fat. (**A**–**C**) Subcutaneous fat, perirenal fat, and brown fat were isolated and weighed. Statistical analysis was performed by Student’s *t*-test (mean ± SEM, *n* = 8,) *** *p* < 0.005. (**D**) White fat and brown fat were examined by H&E staining at (20×) and representative images of gross brown fat are shown. (**E**–**G**) Brown fat was immuno-stained with UCP1 and PPARγ antibodies, respectively. The stained slides were quantified by ImageJ software and the significance was tested by Student’s *t*-test (mean ± SEM, *n* = 3). * *p* < 0.05, *** *p* < 0.005.

**Figure 3 ijms-23-05209-f003:**
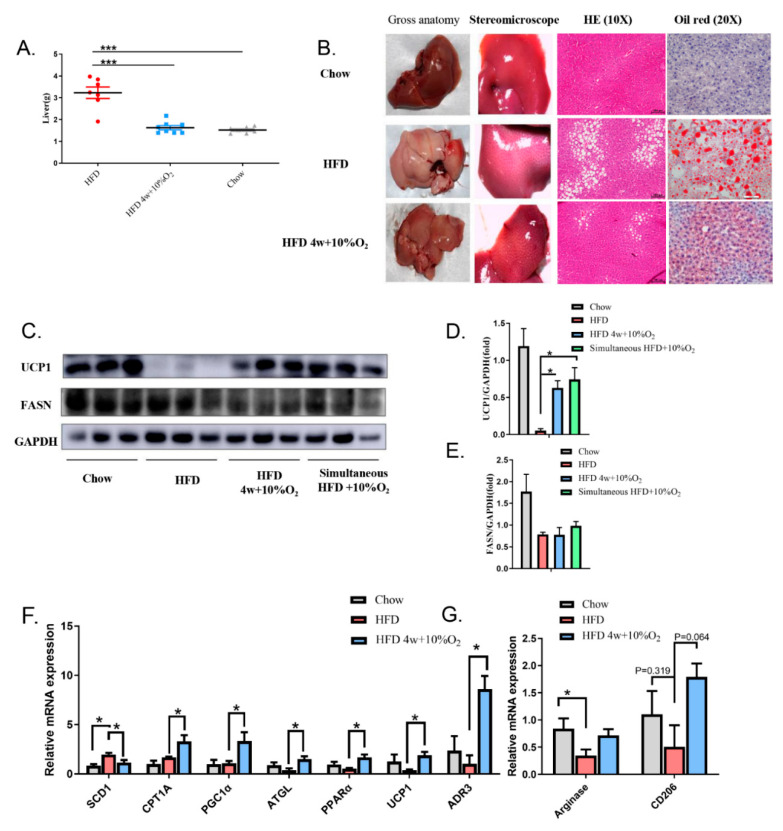
Hypoxia alleviates fatty liver and alters the expression of fatty liver markers. (**A**) Livers were isolated from indicated groups of mice and weighed (mean ± SEM, *n* = 8). *** *p* < 0.005. (**B**) Representative images of gross liver, H&E (10×), and Oil Red O staining (20×) are presented. (**C**–**E**) Liver extracts were blotted with antibodies against UCP1 and FASN. Bands were quantified by ImageJ and normalized by GAPDH. (**F**,**G**) Expression of selected genes for (**F**) lipid metabolism, mitogenesis, and thermogenesis, and (**G**) M2 markers for macrophages in the liver were examined by qPCR, and the signals were normalized by *18S rRNA*. Significance was tested by Student’s *t*-test (mean ± SEM, *n* = 3), * *p* < 0.05.

**Figure 4 ijms-23-05209-f004:**
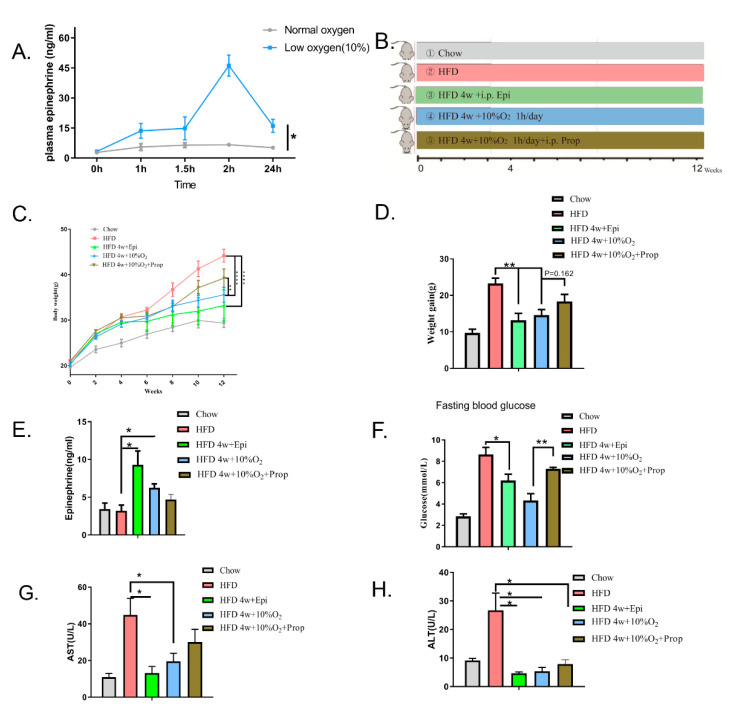
The role of epinephrine in hypoxia-induced changes. (**A**) Chow diet-fed mice were subjected to hypoxia or normoxia for one hour (starting from 0 to 1 h), and serum levels of epinephrine were measured using the ELISA kit. Significance was assessed by two-way ANOVA (mean ± SEM, *n* = 3). * *p* < 0.05. (**B**) Scheme of grouping. Mice were divided into 5 groups (*n* = 8 per group). (1) Chow diet and normoxia, (2) HFD and normoxia, (3) HFD for 4 weeks and then i.p. injection of epinephrine (0.1 mg/kg/day), (4) HFD for 4 w and then 10% oxygen applied, (5) HFD 4 w and then 10% oxygen applied and propranolol (Prop) (2 mg/kg/day) i.p. injected. (**C**) Changes in bodyweight were recorded and statistical analysis performed using two-way ANOVA (mean ± SEM). ** *p* < 0.01. **** *p* < 0.001. (**D**) The value of weight gain was obtained by subtracting bodyweight at the starting point from the endpoint (mean ± SEM, *n* = 8, *t*-test). ** *p* < 0.01. (**E**) Serum concentrations of epinephrine at the endpoint (mean ± SEM, *n* = 5, *t*-test). * *p* < 0.05. (**F**) Fasting blood glucose in mice (mean ± SEM, *n* = 8, *t*-test). * *p* < 0.05. ** *p* < 0.01. (**G**,**H**) Serum levels of ALT and AST at the end of experiments (mean ± SEM, *n* = 5, *t*-test). * *p* < 0.05.

**Figure 5 ijms-23-05209-f005:**
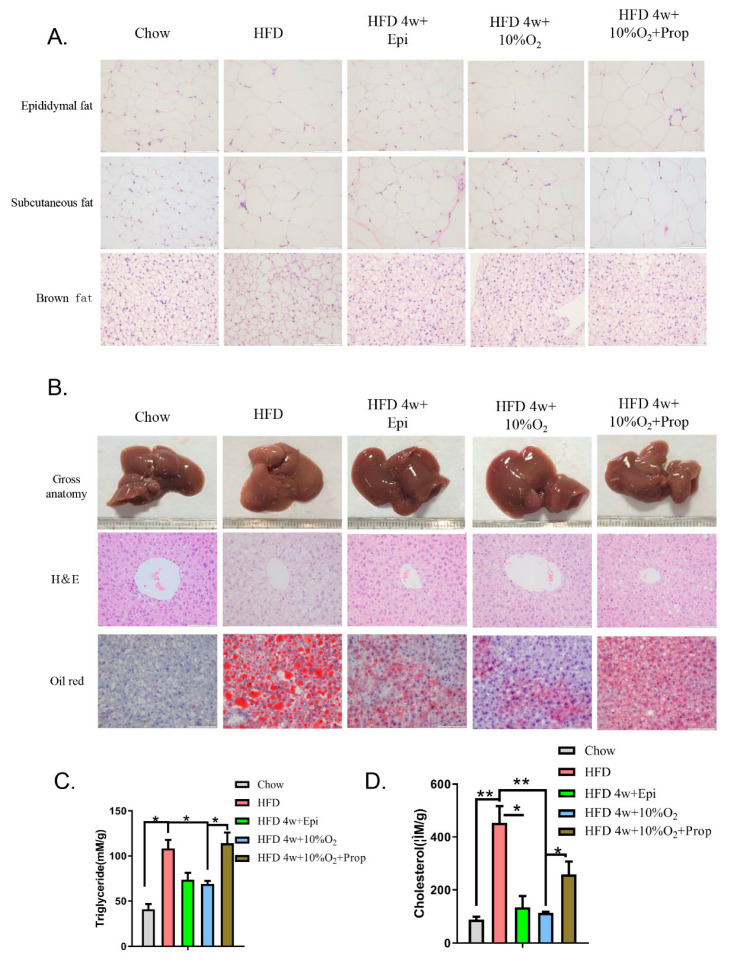
Epinephrine mediates the effect of hypoxia on fat deposits in liver and adipose tissues. Mice were treated as in Figure 4. (**A**) Epididymal fat, subcutaneous fat, and brown fat were isolated and stained with H&E. Representative images are presented (40×). (**B**) Representative images of gross liver and H&E and oil red O-stained liver tissues (40×) are presented. (**C**,**D**) Liver extracts were prepared, and triglyceride (**C**) and general cholesterol (**D**) measured. Statistical analysis was performed using Student’s *t*-test (mean ± SEM, *n* = 3). * *p* < 0.05, ** *p* < 0.01.

**Figure 6 ijms-23-05209-f006:**
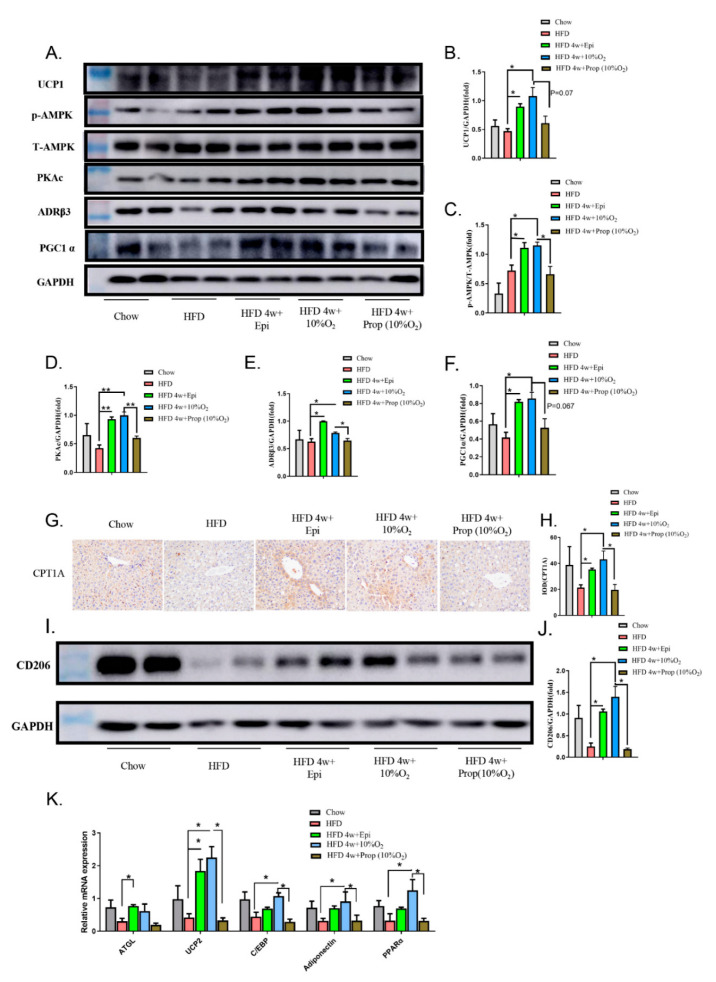
Epinephrine mediates the effect of hypoxia on the expression of modulators for metabolism in liver. (**A**) Liver extracts were blotted with antibodies against UCP1, ADR3, PKAc, P-AMPK, AMPK, and PGC1α. A representative set of blots from two independent experiments is presented. (**B**–**F**) The protein bands in (**A**) were quantified using Image J software. (**G**,**H**) Liver tissues were examined by immunohistochemical staining of CPT1A (40×) and the positive intensity was analyzed by Image J software (**H**). (**I**,**J**) Western blot analysis of CD206 was performed (**I**) and semi-quantification normalized with GAPDH (**J**). (**K**) qPCR analysis was performed on mRNA for *ATGL*, *UCP2*, *C/EBP*, *Adiponectin*, and *PPARα* using *18S rRNA* as an internal control. Student’s *t*-test was used for all statistical analysis (mean ± SEM). * *p* < 0.05, ** *p* < 0.01.

## Data Availability

The data that support the findings of this study are available from the corresponding author upon reasonable request.

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
