# Peer review of "Chronic Intermittent Hypoxia Exposure Alternative to Exercise Alleviates High-Fat-Diet-Induced Obesity and Fatty Liver"

_ijms, 2022, doi:10.3390/ijms23095209_

Round 1
Reviewer 1 Report
The authors describe hypoxia as a new and interesting approach to counteract and reduce obesity and fatty liver, as an alternative to exercise. My major concerns relate to:
1) the use of GAPDH. Hypoxia has been shown to regulate the expression of GAPDH ( PMID: 10364451) and cannot be used as a housekeeper. The authors need to qualify all the experiments with a different housekeeper that is not altered by treatment.
2) The authors should demonstrate that hypoxic treatment triggers activation of the HIF-1a pathway, the master regulator of the hypoxic response analyzing the serologic level of Erythropoietin (Epo).
Author Response
The authors describe hypoxia as a new and interesting approach to counteract and reduce obesity and fatty liver, as an alternative to exercise. My major concerns relate to:
1) the use of GAPDH. Hypoxia has been shown to regulate the expression of GAPDH ( PMID: 10364451) and cannot be used as a housekeeper. The authors need to qualify all the experiments with a different housekeeper that is not altered by treatment.
Response: We appreciate this reviewer’s skepticism about reliability of using GAPDH as an internal control. Because of that, I have searched publications. Our arguments are summarized as follows: first, the article (PMID: 10364451) used extreme condition of hypoxia (1%O2, 24h) and cultured cell lines, which is different from ours (10% O2, 1h); Second, there are many hypoxic studies using GAPDH or Actin as internal controls, for example, Liu et al, Circulation. 2016;134:405–421, Zhang et al, Circulation. 2022;145:1154–1168. Actually, in majority of our experiments, we found that the Western blot signals are consistent with total protein loading amounts, except variations in individual animals, indicating that GAPDH is not affected by oxygen concentration in our settings.
2) The authors should demonstrate that hypoxic treatment triggers activation of the HIF-1a pathway, the master regulator of the hypoxic response analyzing the serologic level of Erythropoietin (Epo).
Response: This is a very good comment and suggestion. The role of HIF-1α in hypoxia-induced changes in energy homeostasis is controversial. On the one hand, HIF-1α is upregulated under hypoxic conditions leading to increased transcription of genes required for glucose utilization (e.g. Glut1, Glut4, phospho-fructose kinase, hexokinase and lactate dehydrogenase, etc). Thus, intermittent hypoxic training or exercise in high altitude hypoxia increases expression of HIF-1α (PMID:11408428, 11581327), suggesting that HIF-1α is beneficial to adaptation to hypoxia by upregulation of gene expression for glycose utilization. On the other hand, genetic studies by hepatic deletion of HIF-1α alleles in mice modeling obstructive sleep apnea have revealed that HIF-1α contributes to insulin resistance and NAFLD (PMID: 34003729). We examined the expression of HIF-1α in fatty liver. As shown in the following, we observed an increase in the liver of HFD fed mice, which was reduced after hypoxia. It appeared that the reduction was more evident if hypoxia and HFD were simultaneously administered. We do not understand the reason behind this phenomenon. Although the underlying mechanism and biological relevance are not clear, we believe HFD itself induces HIF-1α which might in turn plays a role in NAFLD. The reversion of HIF-1α might be accountable to systemic effect of hypoxia. This finding is surprising and interesting albert preliminary. We will further follow up and vigorously assess the role of HIF-1α in the amelioration of NAFLD by hypoxia. We added existing views on the role of HIF-1α in insulin resistance and NAFLD to the Discussion.

Reviewer 2 Report
In this study Yunfei LUO et al test the effect of intermittent hypoxia on the diet induced obesity and fatty liver in mice.
They shown that intermittent hypoxia improves body weight and adiposity in white adipose tissues and liver in a food intake independent manner by increasing brown fat thermogenesis.
They propose that epinephrine is the systemic mediator of this effect that induces the regulation of different markers of lipolysis and thermogenesis in peripheral tissues.
Overall, the manuscript is original and interesting, well written and logically structured.
I have only some concerns that refers to methodology and presentation of the results.
In figure 1F and G: The differences between hypoxia treatments and HFD group are not very visual or intuitive. The authors should add in figure 1 the AUC of this GTT and ITT.
Please increases the size of the figures.
Usually, the weight of the brown fat increases when this tissue is more metabolically active. However, the weights the brown fat in your study in figure 2C is lower in chow compared to HFD. Should be less in HFD. Please explain.
Do you measure possible harmful effects of hypoxia?? Maybe could be interesting to study apoptosis from the liver as a possible secondary effect of hypoxia.
I suggest the authors that they make an effort to explain more in detail the model of hypoxia: correlation with high altitude, the % of O2 in normoxia…etc
Why in fig 2 you do not show the levels of UCP-1 in BAT from all groups I mean could be interesting to show the group of HFD simultaneously with 10% of O2 as well.
The authors should explain the election of the group HFD 4 weeks + 10% hypoxia in all experiments from fig 2 and discard the use of HFD simultaneously 10% O2.
The results concerning fig 1 and fig 4 are not consistent. In fig 1 the fasting glucose is not significant between the control group and 4w+10%O2 but in fig 4 it is. Could you please explain why?
Some references are not properly cited for example ref #13 they do not use hypoxia they perform exercise training.” It has been reported that hypoxia decreases blood levels of glucose, cholesterol and 328 basal Leptin, results in fat loss, and prevents steatosis in obese mice[13-15]”.
“In addition to altitude preacclimatization, IHT has been explored for treatment of a variety of clinical disorders, such as chronic lung diseases, bronchial asthma, hypertension, diabetes mellitus, Parkinson’s disease” lines 318-319. You should cite these articles.
I suggest that the authors add some more references and explain better the mechanism of UCP-1 in liver disease. This is not a common measurement and is a kind of novel mechanism.
- i.e. PMID: 15677488, PMID: 32179129
Another article that worth to mention and that support your results is: PMID: 20134417
My last suggestion is that the authors make some measurements such as Cholesterol, TG and NEFAS from serum in addition to leptin levels to add the ones done in Figure 4.
This could improve the mechanistically part of the article.
Author Response
In figure 1F and G: The differences between hypoxia treatments and HFD group are not very visual or intuitive. The authors should add in figure 1 the AUC of this GTT and ITT.
Please increases the size of the figures.
Response: We added AUC values to Figure 1 legend. Unfortunately, the journal requires insertion of figures in the text body and does not allow us to separately upload figures, which limits the size of the figures. But anyway, we increased the size in our best attempt.
Usually, the weight of the brown fat increases when this tissue is more metabolically active. However, the weights the brown fat in your study in figure 2C is lower in chow compared to HFD. Should be less in HFD. Please explain.
Response: In HFD group, brown fat tends to "whiten", resulting in higher weight.
Do you measure possible harmful effects of hypoxia? Maybe could be interesting to study apoptosis from the liver as a possible secondary effect of hypoxia.
Response: This is a very good point. We performed H&E staining of heart, lung, kidney and even liver tissues and did not find obvious abnormalities in morphology. Here we just show the images of heart, lung and kidney. The liver was ready shown as images or decreases in ALT and AST in manuscript.
I suggest the authors that they make an effort to explain more in detail the model of hypoxia: correlation with high altitude, the % of O2 in normoxia…etc
Response: According to Altitude and Oxygen Chart (https://www.higherpeak.com/altitudechart.html), we chose 10%O2 (~50% normoxia), which is equivalent to 5791 meters (approximately equal to the altitude of Kilimanjaro). Of note, the condition used in our study is isobaric which is easier tolerable than at Kilimanjaro where it is hypobaric. There is a review article drawing a chart on altitude and oxygen pressure (PMID:9774298). Now we briefly added rationale to the Method for using the oxygen concentration.
Why in fig 2 you do not show the levels of UCP-1 in BAT from all groups I mean could be interesting to show the group of HFD simultaneously with 10% of O2 as well.
Response: Because too many groups were hard to monitor the quality of experiments and Figure 1 showed no differences in bodyweight between simultaneously treated group and subsequently treated group, we skipped the simultaneously treated group.
The authors should explain the election of the group HFD 4 weeks + 10% hypoxia in all experiments from fig 2 and discard the use of HFD simultaneously 10% O2.
Response:Please refer to response above. And another reason is that the HFD 4 weeks + 10% hypoxia group is more meaningful in the sense of treatment because people are obese or overweight first. Thus, the results with HFD 4 weeks + 10% would be more meaningful to translation to the treatment of obesity.
The results concerning fig 1 and fig 4 are not consistent. In fig 1 the fasting glucose is not significant between the control group and 4w+10%O2 but in fig 4 it is. Could you please explain why?
Response: The finding of non-significance in Figure 1 probably reflects the size of samples, but it shows a similar trend although not significant.
Some references are not properly cited for example ref #13 they do not use hypoxia they perform exercise training.” It has been reported that hypoxia decreases blood levels of glucose, cholesterol and 328 basal Leptin, results in fat loss, and prevents steatosis in obese mice[13-15]”.
Response: Corrected. We removed ref 13 and added PMID: 20134417, PMID: 15677488, PMID: 32179129
“In addition to altitude preacclimatization, IHT has been explored for treatment of a variety of clinical disorders, such as chronic lung diseases, bronchial asthma, hypertension, diabetes mellitus, Parkinson’s disease” lines 318-319. You should cite these articles.
Response: Sorry. We missed it because it was cited in preceding sentence. Now we added (previous ref 11)
I suggest that the authors add some more references and explain better the mechanism of UCP-1 in liver disease. This is not a common measurement and is a kind of novel mechanism.
- e. PMID: 15677488, PMID: 32179129
Response: We appreciate giving us the references. We added a few sentences in the Discussion
Another article that worth to mention and that support your results is: PMID: 20134417
Response:Appreciate! We added to our references.
My last suggestion is that the authors make some measurements such as Cholesterol, TG and NEFAS from serum in addition to leptin levels to add the ones done in Figure 4.
Response: it is a very good suggestion. Actually, we measured serum levels of cholesterol and triglycerides and mRNA levels of leptin in the liver. We could see improved profiles of these parameters by hypoxia treatment, but not significant. Again, they suggest that we should expand the sample size. Because of the limitation of time for resubmission of this manuscript and the focus of this manuscript, w would leave related tests in future study using animals with specific knockout of AMPK and HIF-1α.

Round 2
Reviewer 1 Report
I would like to thank the author for the responses, but using a HIF-1a target gene as a housekeeper is not technically correct, even if you did not find a significant difference. As you reported in response 2 to my comment, "we believe that HFD itself induces HIF-1α," which supports the possible activation of its target genes. Indeed, GAPDH protein levels are higher in Figure 3C than in the control group.
Author Response
I would like to thank the author for the responses, but using a HIF-1a target gene as a housekeeper is not technically correct, even if you did not find a significant difference. As you reported in response 2 to my comment, "we believe that HFD itself induces HIF-1α," which supports the possible activation of its target genes. Indeed, GAPDH protein levels are higher in Figure 3C than in the control group.
Response: Many thanks for the patience. In our manuscript, there were reference molecules for Western blot and qPCR. According to this reviewer’s suggestion, we conducted experiments again using left-over cDNA and 18S rRNA primers as internal references, which are available in our lab (Figure 3F, G; Figure 6K). As shown in the figures, using 18S rRNA as an internal reference did not change P values albeit a little bit changes in absolute values. Because the university does not allow package delivery due to current situation of Covid-19, we could purchase 28S rRNA primers for amended experiments. As for Western blot references, we could not redo all experiments using a different reference protein as it will take 12 weeks to incubate animals. However, we admit that this should not be a strong excuse if it is necessary. Our reason is that all experiments were performed in duplicates/triplicates or three times on which quantitative analysis was built and that it is not surprising if we see one or two bands were less or more dense than other lanes because we handled animals that intrinsically have variation. We hope the review could accept our reason.
Reviewer 2 Report
Accept
Author Response
Many thanks.
Round 3
Reviewer 1 Report
Thanks for your response.